# A Methodological Review on Development of Crack Healing Technologies of Asphalt Pavement

Lei Zhang [1,*], Inge Hoff [1], Xuemei Zhang [1], Jianan Liu [1,2], Chao Yang [3] and Fusong Wang [4]

1    Department of Civil and Environmental Engineering, Norwegian University of Science and Technology, Høgskoleringen 7A, 7491 Trondheim, Trøndelag, Norway; inge.hoff@ntnu.no (I.H.); xuemei.zhang@ntnu.no (X.Z.); jnliu@chd.edu.cn (J.L.)
2    School of Materials Science and Engineering, Chang'an University, Xi'an 710061, China
3    State Key Laboratory of Silicate Materials for Architectures, Wuhan University of Technology, Luoshi Road 122, Wuhan 430070, China; hbyangc@whut.edu.cn
4    School of Civil and Hydraulic Engineering, Huazhong University of Science and Technology, Wuhan 430074, China; wangfs@hust.edu.cn
*    Correspondence: lei.zhang@ntnu.no

**Abstract:** Crack healing has been a key area of asphalt pavement research. In this review, different crack-healing theories and crack-healing evaluation methods in bitumen and asphalt mixtures are summarized and presented. Then different crack healing technologies have highlighted the problems and solutions associated with their implementation. Detailly, traditional technologies (hot pouring and fog seal) are introduced. They mainly fill cracks from the outside, which can effectively prevent further damage to the asphalt pavement, when the cracks have generally developed to the middle and late stages of practical engineering. Their extension of the life of the asphalt pavement is relatively limited. Energy supply technologies (induction and microwave heating) have demonstrated significant efficacy in enhancing the crack healing capability of asphalt pavement, particularly in microcracks. Now, Extensive laboratory testing and some field test sections have been conducted and they are waiting for the promotion from the industry. The agents encapsulated technologies (Saturated porous aggregates encapsulate rejuvenators, Core-shell polymeric microcapsules, Ca-alginate capsule, Hollow fibers and Compartment fibers) not only heal cracks but rejuvenate the aged asphalt pavement. In order to promote industrial application, more field test sections and large industrial mixing and compaction equipment applications need to be implemented. Finally, some other potential crack healing techniques (coupling application, electrical conductivity, 3D printing, and modifications) are also mentioned.

**Keywords:** asphalt pavement; cracks; self-healing technologies; induction and microwave heating; encapsulated agents

## 1. Background

After thousands of years of development, the road has experienced the original dirt road, flagstone road, gravel road, cement concrete road, and asphalt road. Asphalt pavement consists of an asphalt mixture surface layer, base layer and subbase layer. The asphalt layer is formed by mixing and compacting bitumen, aggregates, filler, and other additives at high temperatures [1]. Asphalt pavement is now the most common surface of roads especially in high-grade road because of its irreplaceable advantages, such as comfort, safety, and low noise.

However, asphalt pavement is also prone to aging and cracking. Water can also enter through cracks and further damage the pavement's base layer [2]. Continuous maintenance is therefore required to ensure quality. And infrastructure related to road construction has basically become the largest expenditure and energy consumption area in most countries and regions [3,4]. More research focused on finding technological solutions to enhance

the reliability and sustainability of pavement materials [5]. Long-lifespan, energy-saving, and functional asphalt pavement are continued research and become the main direction of road development [6]. Crack healing technologies can effectively prolong the service life of asphalt pavement, thereby further reducing the damage to the environment caused by road engineering, reducing energy consumption and $CO_2$ emissions [7]. Making asphalt pavements more environmentally sustainable. Most asphalt pavement distresses and failures start from cracks (fatigue cracks, reflected cracks and low-temperature cracks), so healing cracks in earlier stage can effectively extend serves lifespan of pavement [8]. Although asphalt binder is a kind of self-healing material which can heal spontaneously under special conditions, it is difficult for common asphalt pavement to ensure a heal rate faster than the crack rate in a practical application environment including aging, temperature, traffic loading and moisture [9]. This is mainly because asphalt molecules are hard to attain sufficient moving speed or time to healing cracks by a series of healing theories, including: surface energy theory, molecular diffusion theory and flow behavior and so on [10]. Modifying, external energy supplementary and additives all become methods to accelerate molecular motion and enhance the crack-healing capability of asphalt pavement [11].

Based on the above concepts, many self-healing technologies have been developed in past decades, and some of them have been used in practical engineering field and played significant role until present [12]. Undeniably, fog sealing and pouring asphalt binder are most traditional and universal methods to heal crack in the current actual road maintenance. However, both of them are applied when the cracks have developed for a period of time and macroscopically visible pavement distresses have appeared. They are generally expensive and extended pavement life is limited. Induction heat technology and microwave heating technology can quickly heal cracks by heating asphalt concrete to reduce the viscosity of asphalt binder based on the temperature sensitivity of asphalt binder [13]. Metallic additives are necessary for induction heating, and microwave heating is also very inefficient if no additional absorbing material is added. Rejuvenator encapsulated technology is to encapsulate the agents in capsules or fibres, and then they will be mixed into bitumen or asphalt mixtures. The agents will release and flow into the crack to soften the binder and promote flow, resulting in healing the crack. Once the crack occurs, the capsules or fibres will break. This will be the most effective way to prolong the life of the road. The key point is to control the release time of capsules, which need to meet the bitumen ageing process. Otherwise, it will affect the permanent deformation resistance of asphalt pavement when the capsule is released early. It also solves the shortcomings of difficult penetration and uneven rejuvenation compared with spraying rejuvenator on the surface. There have been many different types of capsule morphologies developed, including saturated porous aggregates that encapsulate rejuvenators, core-shell polymeric microcapsules, hollow fibres, vascular fibres, and multi-cavity Ca-alginate capsules [14]. Apart from the above-mentioned popular technologies, researchers also try other methods to heal cracks. Combination of two or more of the above technologies to improve effectiveness, application of modified binder, and electrically conductive asphalt concrete are proposed to heal cracks. In this paper, asphalt self-healing theories and self-healing evaluation methods are summarized and presented first. Subsequently, various crack healing technologies are introduced, addressing the challenges and proposing solutions related to their implementation.

## 2. Crack Healing Theories

The macroscopic manifestation of crack healing is that the generated crack gradually disappears, and the interfaces on both sides of the crack are integrated until the interface disappears. In this process, a series of physicochemical reactions will occur inside the material, accompanied by the migration of substances and the change of energy [10]. Researchers described the crack healing process in asphalt materials from different perspectives and developed some asphalt crack healing theories based on asphalt's own properties.The Molecular Diffusion Healing Model presents the diffusion process of asphalt molecules

on both sides of the interface with each other until the interface disappears on the basis of polymer chain dynamics [15,16]. Bitumen is a kind of liquid material composed of polymer molecules of different sizes, and its viscosity is sensitive to temperature. Wool et al. proposed that the crack healing process of polymers from the perspective of molecular diffusion consists of five consecutive stages: surface rearrangement, surface approach, wetting, diffusion, and randomization [15,17]. During this process, the mechanical force will be rebuilt due to the secondary bonds being restored among molecules or microstructural components by Rouse diffusion or reptation [18]. In addition to the chemical molecular composition of the asphalt material itself, the two most important factors that determine crack healing are temperature and time. In brief, a higher temperature response leads to a shorter healing time. Sun et al. proposed a recovery function of asphalt binder based on the molecular diffusion by fatigue-rest-fatigue test, which can consider the effect of healing time and temperature [19], shown in Equation (1).

$$HI(t, T) = HI_0(T) + D_0 \exp\left(-\frac{E_h}{RT}\right) \cdot t^{0.25} \tag{1}$$

where $HI(t, T)$ is the crack healing ratio. $HI_0(T)$ is instant healing ratio. $D_0$ is a diffusion parameter, $R$ is the universal gas constant (8.314 J/mol/K), $E_h$ is the activation energy. The function adds the material parameter, activation energy, which is determined by the chemical composition of the material itself. It also can define the self-healing potential of asphalt binder. Zhang et al. tested the self-heal capability and calculated the activation energy of different aged levels asphalt binder, resulted in self-healing capability and activation energy both decrease with ageing increases [20].

Phase field healing theory describes that the separated phases will be reconfigured and tend to be isotropic during the heating process. The phases on both sides of the crack are unevenly distributed. The motive force of its driving is that in the process of mixing fluid substances, it always tends to the minimum entropy to move. The mechanical properties of asphalt can be restored when the temperature is reduced [21]. Small molecules and long-chain polymeric molecules are not common to be found in asphalt, and the diffusion model doesn't describe all the crack healing behavior completely. Kringos et al. developed the phase field healing theory by observing the phase movement of asphalt surfaces by AFM (atomic force microscopy) investigations [22]. Other researchers also observed the two phases getting separated and reaching equilibrium by AFM images [23]. Microcracks usually occur at the interfaces between the phases due to stress concentrations. The phase will reconfigure to a new homogeneous mix once the thermodynamic conditions change due to energy application.The surface energy healing theory presents that the decrease of crack surface energy during the crack-healing process of asphalt. Simply put, calculating the energy required due to the reduction in surface area. Lytton et al. derived the energy balance of the crack interface during the disappearance process from fracture mechanics during the fatigue process of viscoelastic materials [24], shown in equations below.

$$2\Gamma_h = E_R D_h(t_a) H_v \tag{2}$$

$$D_h(t) = D_{0h} + D_{1h} t^{mh} \tag{3}$$

$$2\Gamma_f = E_R D_f(t_a) J_v \tag{4}$$

where $\Gamma_f$ is the fracture surface energy density, $\Gamma_h$ is healing surface energy density of a crack surface, $E_R$ is the reference modulus, $D_f(t_\alpha)$ is the tensile creep compliance, $D_h(t_\alpha)$ is compressive creep compliance, $J_v$ is the integral of $J$, and $H_v$ is $H$ integral of $H$. $m_h$ is the slope of creep compliance of concrete with log time.

Through the derivation of the above equations, Si et al. defined two healing rates ($h_1$, $h_2$) by different mechanisms and fatigue cracking processes for vicoelastic medium derived by Schapery [25,26]. $h_1$ is the short-term healing rate decided by non-polars; it usually occurs quickly. $h_2$ is the long-term healing rate determined by polar; it is time-dependent. The total healing $h$ is shown in the below equations.

$$h = h_2 + \frac{h_1 - h_2}{1 + \frac{h_1 - h_2}{h_\beta}(\Delta t)_h} \tag{5}$$

$$h_2 = [\frac{2r_m E_R^2 D_{1h} \Gamma^{AB}}{(1 - v^2) C_m^{1/mh} H_v}]\beta \tag{6}$$

$$h_1 = [\frac{K_h E_R D_{1h} H_v}{2\Gamma^{LW}}]^{\frac{1}{mh}}\beta \tag{7}$$

where $(\Delta t)h$ is the rest period, and $h\beta$ is maximum healing ratio of bitumen

Si et al. investigated healing rate of 12 kinds of asphalt concrete, it was found that short-term healing rate ($h_1$) had significant difference while the long-term healing rate ($h_2$) were relatively same [26].

The capillary flow healing theory states that the asphalt binder can flow and fill an open crack automatically due to capillary force, based on the fact that asphalt is a kind of liquid material. When the crack growth reaches a certain size, molecular diffusion cannot occur through a large gap. The healing of the cracks can also be observed at this time, mainly because the liquid bitumen fills the cracks under capillary pressure. The capillary flow healing theory can present healing efficiency through a modification of the Lucas-Washburn equation [27]. It can be found that increasing the surface tension force of asphalt can effectively promote the flow of asphalt into cracks. As the temperature increases, the viscosity of the asphalt decreases, and its surface tension also increases. This healing therefore occurs at relatively high temperatures or when the viscosity of the asphalt interface decreases due to infiltration by the rejuvenator.

## 3. Evaluation Methods of Crack Healing

Several evaluation methods have been used to quantify the crack healing level (HL) by measuring the performance of asphalt before and after the healing process. Table 1 summarises the evaluation methods for asphalt binder and mixture. It is found that the healing evaluation of bitumen is based on its own performance recovery by material properties, like complex modulus, dissipated energy, fatigue life, etc. These are mainly due to changes in the microstructure and uneven phase distribution in the bitumen under loading conditions. After a period of rest, bitumen becomes a homogeneous mix again through molecular diffusion and phase field reconfiguration, resulting in performance recovery. As for mixture, crack healing level (HL) is based on mechanical property recovery, mainly because this meets the pavement design theories of mechanical property attenuation and distress.

**Table 1.** Self-healing evaluation methods on asphalt binder and mixture.

| Materials Types | Test Method | Healing Parameter | Healing Indicator | Notes | Ref. |
|---|---|---|---|---|---|
| Binders | Ductility | Ductility value | $HI = \frac{L_{healed}}{L_{original}}$ | $L_{healed}$ and $L_{original}$ are ductility test result before and after break-healing | Qiu, J. et al. [28] |
| | DSR sweep test | Complex modulus and number of cycles | $HI = 100 \cdot \frac{G*_{Terminal}}{G*_{initial}} \frac{N_{after} - N_{before}}{N_{before}}$ | $G_{initial}$ and $G_{terminal}$ are the dynamic modulus before and after loading test; $N_{before}$ and $N_{after}$ are the numbers of cycles before and after rest period; | Tan, Y. et al. [29] |
| | Fatigue-rest-fatigue test using DSR sweep test | Area under the curve | $HI = \frac{A_d}{A_{before}}$ | $A_{before}$ and $A_d$ is the area between the curves of the modulus versus the number of load cycles and the line of $1/2$ modulus before and after rest; | Shan, L. et al. [30] |
| | Fatigue-rest-fatigue test | Complex shear modulus | $HI = 100 \cdot \frac{G*_{h0}}{G*_0}$ | $G*_0$ and $G*_{h0}$ are the complex shear modulus before and after healing; | Qiu, X. et al. [31] |
| | Fatigue-rest-fatigue test | Dissipated energy | $HI = 100 \cdot \frac{W_{after}}{W_{before}} \frac{N_{after}}{N_{before}}$ | $W_{before}$ and $W_{after}$ are the initial dissipative energy before and after healing | Qiu, X. et al. [31] |
| | Fatigue-rest-fatigue test | Fatigue life | $HI = \frac{\sum N_{fo}}{\sum N_{f1}}$ | $\sum N_{fo}$ and $\sum N_{f1}$ are the fatigue life after and before rest | Liu, G. et al. [32] |

**Table 1.** *Cont.*

| Materials Types | Test Method | Healing Parameter | Healing Indicator | Notes | Ref. |
|---|---|---|---|---|---|
| Asphalt mixture | IDT | Resilient modulus | $\text{HI} = \frac{MR(t) - MR0}{MR_{undamaged} - MR0}$ | $MR(t)$ is the normalised resilient modulus at time $t$; $MR0$ is the normalised resilient modulus at $t = 0$; and $MR_{undamaged}$ is the undamaged normalised resilient modulus | Chen, Y. et al. [33] |
| | SCB test or 3-point bending test | Strength | $\text{HI} = \frac{F_{after}}{F_{before}}$ | $F_{after}$ and $F_{before}$ are fracture peak load after and before healing | Riara, M. et al. [34] |
| | SCB test or 3-point bending test | Stiffness | $\text{HI} = \frac{S_{after}}{S_{before}}$ | $S_{after}$ and $S_{before}$ are stiffness after and before healing | Riara, M.et al. [34] |
| | SCB test or 3-point bending test | Fracture energy | $\text{HI} = \frac{E_{after}}{E_{before}}$ | $E_{after}$ and $E_{before}$ are fracture energy after and before healing | Riara, M. et al. [34] |
| | Four-point bending fatigue-healing-fatigue test | Stiffness modulus | $\text{HI} = \frac{S2 - SS}{S1 - SS}$ | $S1$ and $S2$ are the initial stiffness modulus before and after rest; $SS$ is stiffness modulus when beam reaches fatigue condition | Xiang, H. et al. [35] |
| | Four-point bending fatigue-healing-fatigue test | Fatigue life | $\text{HI} = \frac{N_{f-after}}{N_{f-initial}}$ | $N_{f\text{-}after}$ and $N_{f\text{-}initial}$ are fatigue life after and before resting | Liu, Q. et al. [36] |

## 4. Crack Healing Technologies

### 4.1. Hot Pouring

Hot pouring and fog sealing are the most traditional and universal methods to heal cracks in the current road maintenance. Hot pouring is usually applied for big and long structure cracks, while fog sealing is used for fatigue cracks and aged pavement surfaces [37]. The pavement heat pouring technology is mainly used to pour hot polymer materials with strong cohesion and elasticity into the cracks of the asphalt pavement. The polymer material is usually made of base asphalt, high molecular weight polymers, stabilizers, additives, and other materials. The method is used to repair large and long transverse cracks and horizontal cracks [37]. The road can be maintained in time to prevent the damage caused by the entry of rainwater and impurities and the appearance of potholes inside, thus extending its service life [38]. However, it can only be done completely manually, which is expensive and inefficient. And the cracks have extended to the last period when the materials can pot into the cracks. Hot pouring technology can not extend service life too much (Figure 1).

### 4.2. Fog Sealing

A fog seal is an application of asphalt emulsion to an existing asphalt pavement surface that has aged and developed cracks or other distresses (Figure 2). After the asphalt emulsion is demulsified, the fog seal material can penetrate the aged asphalt concrete and also flow into the micro-cracks. It can rejuvenate aged asphalt concrete, close interconnected voids, heal cracks, and prevent moisture damage [39]. The technology could usually extend the surface performance of asphalt pavement for 2–3 years [40].

There are still certain disadvantages to the application process. After the asphalt emulsion breaks and cures, the cracks and voids are closed, and the skid resistance of the asphalt pavement surface will reduce significantly. Traditional fog seal materials face adhesion loss and spalling failure because of low mechanical strength and inadequate cracking resistance [41,42]. Islam's research has shown that spraying fog seal reduces the coefficient of friction of pavements by 20% to 40% [42]. Low diffusion and uneven spraying combined with the decline of anti-skid resistance will not only affect the damage to the pavement structure but also seriously affect the safety of driving. Modified emulsified bitumen with polymer material (like styrene-butadiene-rubber (SBR) or styrene-butadiene-styrene (SBS)), which helps to increase the adhesion of the fog seal material [43]. Xu et al. found that waterborne additives (acrylates, cationic acrylates, and polyurethane) can improve the thermal cracking resistance of SBS-modified asphalt emulsions through the

synergy effect [41]. Liu et al. used epoxy resin to modify fog seal materials and found it could shorten the surface dry time and increase the waterproofing and durability of asphalt pavement [44]. In order to increase the skid resistance performance of the pavement surface after fog sealing, a sand fog seal concept was proposed, which is composed of fog seal materials and fine sand. Ma et al. designed sand fog seal materials with different sand contents and optimised the 25% sand content to balance stability and fluidity well [45]. Zhang et al. also determined the best formulation of sand fog seal mortar and a method for predicting traffic open time based on image processing technology [46].

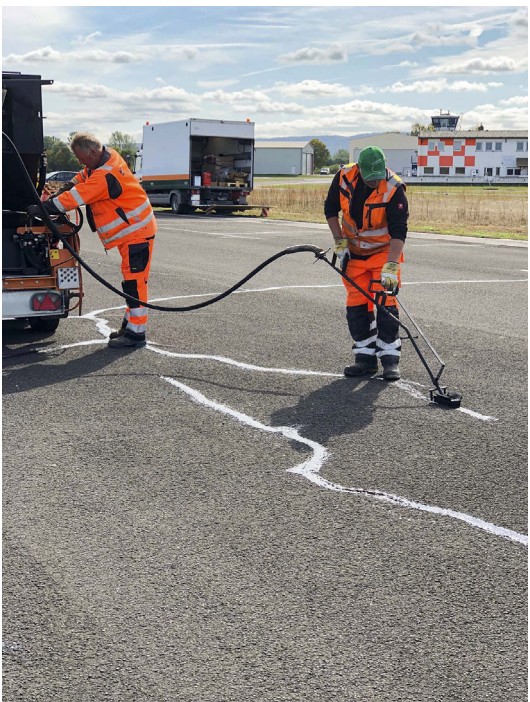

**Figure 1.** Hot pouring technology to heal the cracks.

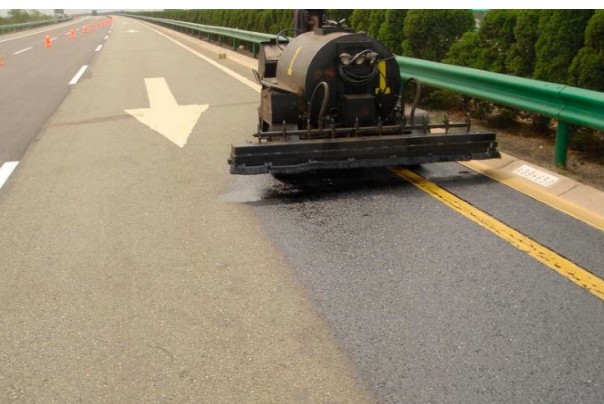

**Figure 2.** Fog seal technology with operation.

*4.3. Heating Technologies*

Bituminous material itself is a self-healing material capable of repairing cracks, but sufficient temperature and time are required. Mainly because the viscosity of bitumen is temperature-sensitive; the higher the temperature, the smaller the viscosity [47]. Heating the bitumen can reduce the viscosity of the asphalt, which not only reduces the surface energy of the crack but also makes the bitumen have better fluidity to flow into the crack and heal it [48]. And high temperatures help improve bitumen molecular diffusion to heal the cracks. Based on the theories, heating asphalt concrete technologies were proposed to

heal the crack that occurred in the service process due to ageing, traffic loading, or fatigue. Induction heating and microwave heating are efficient and non-contact heating methods that are suitable for heating asphalt pavement [49]. They just take dozens of seconds to heat up to asphalt's self-healing temperature.

### 4.3.1. Induction Heating

The basic principles of induction heating technology are electromagnetic induction and joule heating. During the test process, when the material is placed in the induced magnetic field, its interior will experience an electro-motive force according to Faraday's law. Then the eddy currents are generated to heat the materials by Joule's law. Besides, magnetic domain rotation will also produce hysteresis heating when applied to magnetic materials.Bituminous materials themselves cannot be heated directly by induction; some electrically conductive materials (like steel fibres, steel wool, steel slag, etc.) are considered to be added to asphalt mixtures [50,51]. After the conductive material heats up under an induction field, it can be conducted to the bitumen to heat up and heal the cracks [52]. The basic principle is shown in Figure 3. In Garcia et al.'s study, steel wool fibres were added to dense asphalt mixtures. The results show that a higher volume of steel wool fibres in the mixture may increase the air void content of asphalt concrete and affect its thermal conductivity [53]. When Liu et al. incorporated steel fibres, steel wool, and steel slag into porous asphalt concrete, it was found that the conductivity of samples with longer fibres with a smaller diameter was better than that of samples with shorter fibres with a larger diameter [54]. And the steel fibres and steel wool were more beneficial in increasing temperature than steel slag; they also improved the flexural strength of asphalt concrete [55]. 8% steel wool is considered the optimal content and increases the indirect tensile strength, resilient stiffness, and fatigue resistance of porous asphalt concrete [56]. An appropriate amount of steel fibres can increase the basic mechanical properties of asphalt concrete while also increasing its thermal properties.

Induction crack healing technology is mainly used to increase the temperature of asphalt concrete and reduce the viscosity of asphalt to promote crack healing. Temperature is the main index used to determine the crack healing rate [57]. Liu et al. believed that 85 °C was the optimal surface temperature for crack healing through a three-point bending test and that the optimal strength recovery rate could reach 78.8% [58]. Liu et al. designed the four-point blending test to investigate the fatigue life extension of asphalt concrete with steel fibres. Micro-strain amplitudes would affect the size of cracks between samples of the test. Higher micro-strain amplitudes and larger cracks were generated in the fatigue testing, which is hard to heal. Liu et al. found that the fatigue life extension ratio during different micro-strains and temperatures for hot mix asphalt and warm asphalt shows that induction heating technology can expand over 70% of the life by healing the crack [36,59]. Another advantage of induction heating technology is that the asphalt pavement can be heated multiple times during its service life. The strength recovery rate and fatigue life extension of asphalt concrete containing steel fibres are still unchanged after 5 cycles of induction heating [60,61]. Yang et al. also studied the induction heating crack healing level of asphalt concrete containing reclaimed asphalt pavement (RAP) and steel slag. The strength healing rate of RAP containing 40% can still reach 57.9%. At the same time, the crack healing rate will decrease by 10% after four cycles [62,63]. In December 2010 in the Netherlands, the first induction-heating asphalt pavement was paved [61], and it has been applied until now and remained well. In June 2014, this trial section received the first induction heating treatment, and it showed good healing ratios and ravelling resistance [64]. In 2018, in the Guangdong Province of China, a 400-metre induction healing pavement test section was also paved [4]. According to estimates, if all roads in the Netherlands were replaced with induction-heated pavements, it could save approximately 90 million euros every year, and the life span of roads would also extend by 50%. By the same calculation, China will save a maintenance expense of 1000 billion RMB if only 10% of the roads are replaced by induction healing pavemeng [56]. However, induction heating technology cannot prevent the ageing

of asphalt pavement. And the pavement will be more and more prone to cracks, and the healing temperature will be higher than ever during the service process. The total mixing time should reach 5 min to get a homogeneous mix with minimum steel wool clusters. It is six times more than when normal asphalt pavement is applied [56]. And the heating efficiency is also limited; one hour can only heat 5 km since it takes 26.4 s for the temperature of the road surface to increase from 5 to 85 °C [65]. Large-scale induction equipment also requires further study in order to guarantee heating rates and maintenance times.

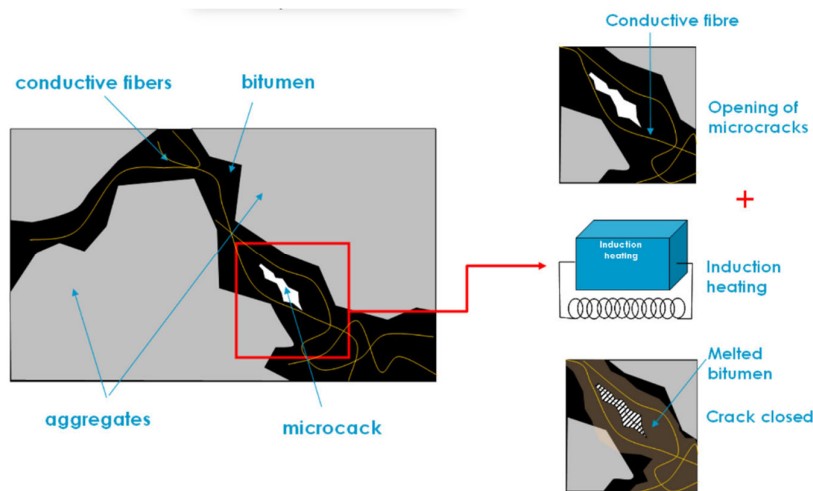

**Figure 3.** Induction heating scheme.

4.3.2. Microwave Heating

Microwave heating is also a good non-contact heating method used in road surface heating. The basic principle is that the direction of the electric polarity molecules in the object will vibrate with the oscillating electric field, and the inherent electromagnetic field of a molecule is changed and affects the adjacent molecules, so the vibration of the molecule is in the molecule passed between under the action of the electromagnetic field oscillated by microwave high frequency. Molecular vibration is internal energy, and increasing internal energy is like heating an object [66]. Microwaves cannot heat non-polar molecules. Bitumen contains a large number of polar molecules. And Lou et al. reported that microwave heating rarely aged asphalt, referring to the ageing index [67]. The mechanism of crack healing is the same as induction heating after the mixture is heated [68]. Researchers found that the incorporation of some additives can improve microwave heating effectiveness. Gallego et al. reported that 0.2% steel wool can significantly improve microwave heating potential, whereas only 1/10 dosage for induction heating can reach the same heat rate [69]. Li et al. found that nanometer microwave-absorber materials (like SiC, CNTs, and graphene) can also improve the microwave heating rate and healing properties of asphalt [70,71]. Sun et al. studied the heating and healing properties of a mixture incorporating steel slag and steel fibres using both induction and microwave technologies [72]. Lou et al. used three kinds of steel slag (hot braised, hot pour, and iron slag) to replace the coarse aggregates in the asphalt mixture and found that the microwave heating rate could improve. And 60% replacement is the most effective dosage [73]. Lou et al. also used ferrite fillers to replace limestone fillers in a mixture that incorporated steel slag, and the microwave heating rate was further improved [74]. Metallic powder and fly ash can also improve the microwave heating potential of asphalt [75,76]. In the Jahanbakhsh et al. study, carbon black increased the heating rate of asphalt pavement made of limestone and siliceous types of aggregate by 47% and 25%, respectively [77]. Wang et al. also found that pyrolysis carbon black (PCB) had good microwave absorbing performance for asphalt pavement. And the addition of 15% PCB increased the self-healing rate of bitumen by 3.59 times [78]. At present, the microwave heating cracking technology in the laboratory mainly relies on microwave ovens. The heating time and temperature are the biggest factors affecting the crack healing

rate of asphalt concrete. Jose et al. comparatively studied the microwave and induction heating, the microwave healing rate of dense asphalt mixtures was superior to the induction healing rate through three-point bending test, the strength recover rate reached 93% for microwave healing, while 75% for induction heating. This is mainly because microwave heating is a holistic heating, while induction heating has a temperature gradient from top to bottom, and the bottom crack does not reach the optimal level of healing [13]. Zhu et al. investigated healing level of asphalt concrete with base bitumen and SBS modified bitumen in different temperature by semicircular bending test, it was found that the strength recover ratio could reach 85% in 80 °C [79]. The microwave heating healing ratios of asphalt mixtures with different structure (semi-dense, porous, and gap-graded) were investigated in Franesqui et al. study. It was found that top-down cracks (<4–5 mm)can be completely healed by microwave heating [75]. Crack healing level of asphalt concrete containing RAP was also investigated by microwave heating, the RAP content adversely affected the microwave healing [80]. Nevertheless, there are obvious disadvantages to microwave heating technology. It is challenging to control the microwave's reflection from the flat surfaces. And the human body can also absorb microwave energy, which can heat exposed tissues and cause thermal damage [81]. Therefore, further extensive research needs to be conducted to enhance the efficiency of microwave healing on asphalt pavements while ensuring safety is not compromised.In heating technology, conductive materials (steel fibres and steel slag) or absorbing microwave materials (ferrite, SiC, CNTs, graphene, fly ash, and carbon black) that are added to asphalt concrete basically belong to the waste products of other industries. Specifically, steel fibres and ferrite come from waste from the steel cutting and forging industries. Steel slag is a waste product after steelmaking [82]. Reusing asphalt concrete can turn waste into treasure, make full use of resources, reduce the capture of new resources, and be more sustainable.

### 4.4. Agents Encapsulated Technology

The agent-encapsulated technology solves the shortcomings of the spraying rejuvenator, such as difficult penetration and uneven distribution. Rejuvenating aged asphalt pavement when the aging starts to result in cracking is the optimal solution. This will be the most effective way to prolong the life of the road. It is crucial to avoid prematurely releasing the rejuvenator before the bitumen has been aged, which will reduce the permanent deformation resistance of the road [83].

#### 4.4.1. Saturated Porous Aggregates Encapsulate Rejuvenators

Garcia et al. first proposed the asphalt self-healing technology with capsules by applying self-healing capsules to cementitious materials [84]. The porous sand is used as the core material, and its internal porous structure can effectively store the rejuvenator. Then, the surface is coated with epoxy resin and cement as a shell. The capsules of various sizes (with diameters ranging from 1.6 to 7.1 mm and shell thicknesses ranging from 0.1 to 0.35 mm) were produced by the same procedure. It is found that larger capsule cores contain more oil, and their force resistance is over 10 N in capsule compression tests [85,86]. The saturated porous aggregate capsules were added to porous asphalt concrete and could resist high temperatures (up to 180 °C) and tension during the mixing and compaction of the asphalt mixture [86]. It was found that the capsules inside the porous concrete could break and the oil could flow out and diffuse into bitumen after the indirect tensile test, and the crack could be effectively healed, as shown in Figure 4 [87]. However, the capsules were unevenly distributed in the asphalt mixture. They tended to accumulate at a certain depth in the test sample. The capsules decrease the indirect tensile strength of asphalt concrete. This happened because its strength is lower than normal aggregates [87]. Oil in the capsule is also extremely limited owing to the fact that most holes in the porous aggregates are closed. This kind of capsule is not particularly suitable for asphalt concrete to improve its self-healing ability.

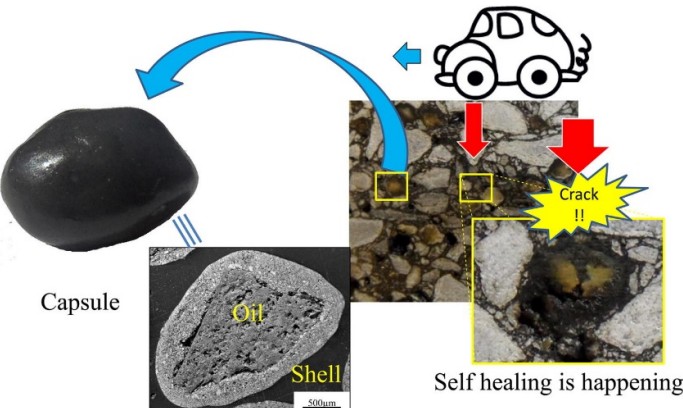

**Figure 4.** Characterization and healing process of saturated porous aggregates encapsulate rejuvenators [87].

### 4.4.2. Core-Shell Polymeric Microcapsules

Core-shell polymeric microcapsules involve two primary components: the core material (rejuvenators) and the shell materials [88]. These capsules are categorised based on their size as nano-capsules (1 m), micro-capsules (1 m < D < 1000 m), and macro-capsules (>1000 m) [89]. The section focuses mainly on the application of microcapsules in asphalt crack healing. The core material may be surrounded by one or two layers of shell materials. There are two types of core-shell capsules: (1) single-shell capsules [90] and (2) double-shell capsules [91]. Through in-situ polymerization and two-step coacervation processes, a type of core-shell microcapsule was created, which significantly reduced the size of encapsulated rejuvenators and improved their incorporation into bituminous materials [92]. These microcapsules consist of a rejuvenator core surrounded by a protective shell, which can be ruptured by propagating crack fronts, allowing for the release of the healing agent through capillary action. TGA tests have indicated that these microcapsules can withstand the storage and mixing temperatures of bitumen [93]. Sun et al. have demonstrated that they can resist mechanical agitation at high temperatures while still being able to release the rejuvenator during loading, thereby enhancing the healing capacity of aged bitumen [94]. Furthermore, fatigue-rest-fatigue tests have shown that the inclusion of microcapsules containing rejuvenator can enhance the fatigue resistance of aged asphalt binder, with a 1 wt% concentration resulting in a 45.68% increase in fatigue life compared to non-aged neat bitumen [90]. At the same time, a commercial prepolymer of melamine-formaldehyde modified by methanol (MMF) was also developed as the shell material of microcapsules [91,92,95,96]. Its SEM images are shown in Figure 5. Thermal tests showed that the microcapsules survived in 200 °C bitumen and could improve the self-healing ability of bitumen with the capsules [97]. Moreover, nano-$CaCO_3$ powder was added to MMF as the shell to enhance adhesion with bitumen and thermal stability; these nano-$CaCO_3$/polymer microcapsules survived in bitumen for a long service time under radical conditions without damage because of their good thermal stability [96].

However, the capsule does have its limitations. Its preparation efficiency in the laboratory is relatively low, and the quantity of each preparation is limited, making it challenging to use on a large scale in road engineering. Additionally, there have been no experiments conducted on adding it to the asphalt mixtures; it is only added to binders. It is difficult to prove that it can also keep itself whole without breaking during mixing and compression. As the capsule has a single core, once it ruptures, all the rejuvenators inside are released completely, which could potentially reduce the asphalt concrete's resistance to permanent deformation.

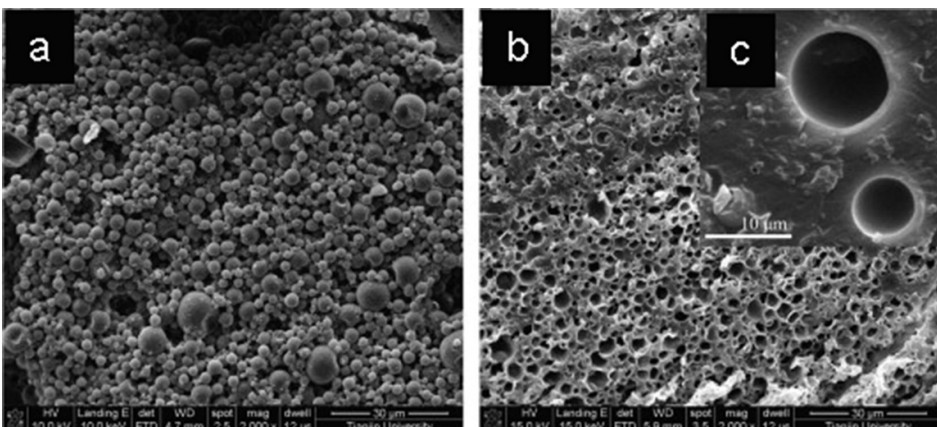

**Figure 5.** SEM images of core–shell microcapsules: (**a**) external aspect, and (**b**,**c**) cross section of the microcapsules [91].

### 4.4.3. Ca-Alginate Capsule

Calcium alginate capsules are prepared by dripping the emulsion containing sodium alginate, surfactant, and regenerant into calcium chloride solution. The shell material of the capsule is to fix the chain alginate ions into a three-dimensional network of calcium alginate material by the method of ion replacement. Therefore, the rejuvenator can be stored in the gaps of the network structure [98]. Rao et al. also innovated the experimental apparatus and successfully fabricated the capsules with industrial raw materials on a large scale in the laboratory [99]. The size of capsules is typically 1.5–3 mm, and the capsule is incorporated into the asphalt concrete as an aggregate. Its internal structure is a multi-chamber structure [100], which enables multiple releases of the internal rejuvenator for long-term crack healing, as shown in Figure 6.

This capsule healing technology uses an internal rejuvenator to soften the asphalt on the crack interface, reduce its surface energy, increase the flow activation energy of the asphalt, and promote crack healing. It is generally believed that there are two release mechanisms for the rejuvenator inside the calcium alginate capsule. The first one is the same as the core-shell capsule. The crack or loading induces the capsule to rupture and releases the rejuvenator inside, which can effectively heal the crack. Because the damaged capsules embedded in asphalt mixtures can be found by CT scan [101,102], as shown in Figure 7, The other is elastic contraction and expansion. Since the chambers inside the capsule are not all closed, under the action of the load, the capsules are deformed by pressure, the internal rejuvenator will flow out, and the capsule does not rupture. The release method could rejuvenate aged bitumen and improve its own self-healing capability [103]. The researchers studied the mechanical and thermal properties of the capsule itself. The uniaxial compression test to make capsules break is usually used to measure the mechanical strength of capsules. It was proved that the capsules could resist the pressure during mixing and compressive forces when the strength was higher than 10 N [84,104]. Zhang et al., Wan et al., Xu et al., Micaelo et al., Al-Mansoori et al., and Norambuena-Contreras et al. tested the strength of this capsule at different temperatures; the results ranged from 12–33 N [101,105–108]. By thermal gravimetric analysis (TGA), the capsules had a mass loss of 2.9–4% at 200 °C and 3.8–5.5% at 160 °C [101,105–108]. The rejuvenator content of the capsule weight could also be calculated by the thermogravimetric curve; it ranged from 52–80% [101,105–108]. It can be seen that the capsule can fully survive the mixing and compression of asphalt concrete.

Norambuena-Contreras et al. Incorporating 0.25–1% capsules into dense asphalt concrete (AC 13), 0.5% capsule content has basically no effect on the basic road performance (scattering, indirect tensile strength, Marshall stability, and freeze-thaw cycles) of asphalt concrete and only slightly reduces its resistance to permanent deformation [100]. Al-Mansoori et al. added the capsules to AC20 asphalt concrete and got similar results.

When the capsules were added to SMA asphalt concrete, its stiffness and deformation resistance decreased and had no influence on fatigue resistance [109]. Xu et al. found the capsules could reinforce the stiffness modulus of porous asphalt concrete.

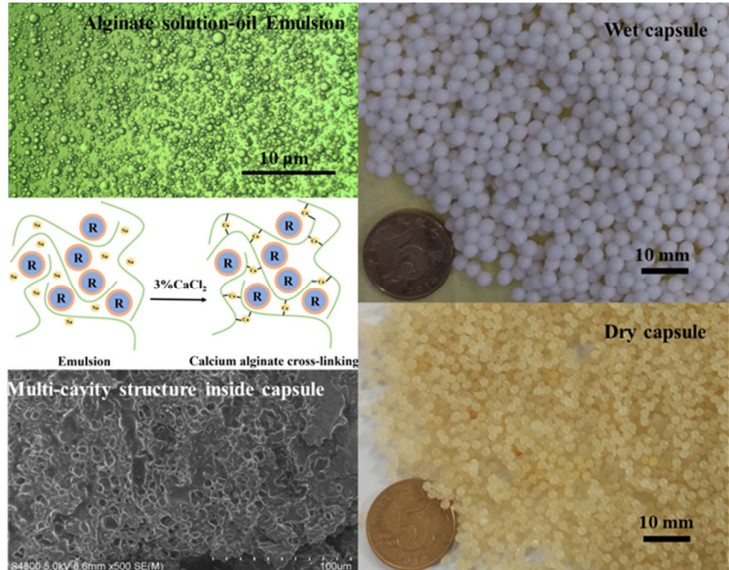

**Figure 6.** Morphological characterization of calcium alginate capsules.

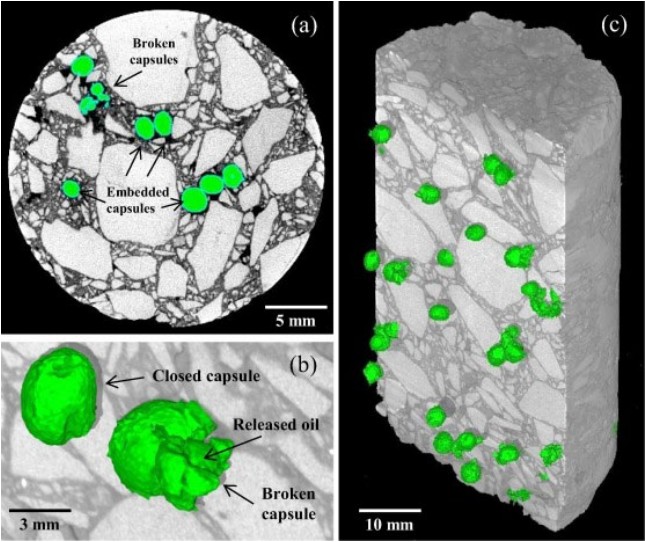

**Figure 7.** CT-Scans reconstructions of the asphalt mixture with capsules, (**a**) sectional view, (**b**) broken capsules in mixtures, (**c**) 3D image [104].

More importantly, this capsule can effectively increase the crack healing ability of asphalt concrete in low temperature. Al-Mansoori et al. found that the capsules can effect the healing levels of samples when the healing temperature below 40 °C, asphalt samples with and without capsules owned same healing levels when temperature was over 40 °C by fracture- rest-healing-refracture test [106]. Zhang et al. found that the strength recovery ratio and fracture energy recovery ratio of asphalt with capsules could reach 92.7% and 180.2% respectively, while the reference samples are 61% and 31.5%, shown in Figure 8. The capsule also significantly improve the healing capability of porous asphalt concrete by semi-circular bending (SCB) test by Xu et al. [110]. The self-healing property of reflective crack of asphalt mixtures with capsules was investigated in Garcia-Hernández et al. study. It was found that the capsules had the best self-healing efficiency in porous asphalt concrete, then

stone mastic asphalt mixture, the last one dense asphalt mixture [111]. Jose et al. also used waste cooking oil as the rejuvenator to synthesis the alginate capsules, it was found that waste cooking oil capsules presented feasibility for self-healing applications for mechanical and thermal stability and physical-chemical properties. The capsules can diffuse in the aged bitumen, reducing its viscosity and promoting the self-healing of microcracks [112].

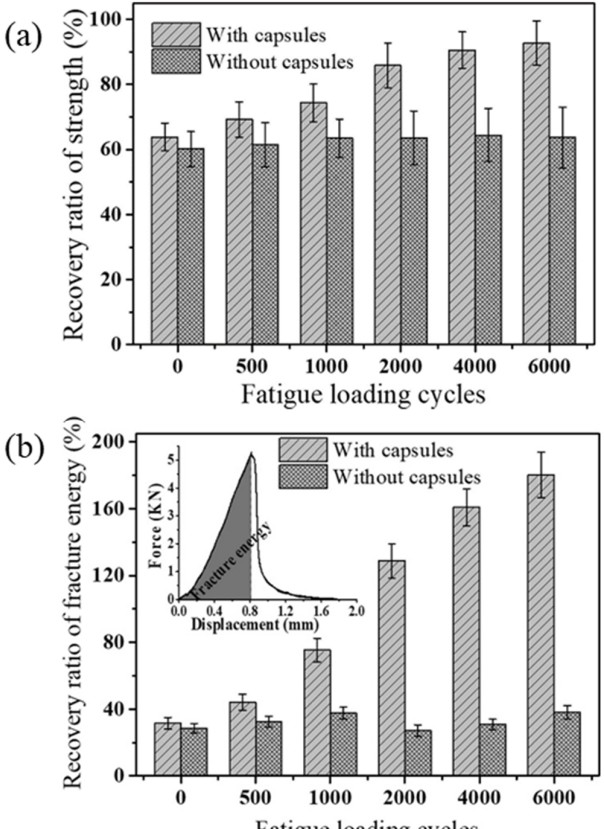

**Figure 8.** Recovery ratio of asphalt mixture with and without capsules: (**a**) strength, (**b**) fracture energy [98].

### 4.4.4. Hollow Fibers

Hollow fibres are used to provide a healing mechanism similar to that of encapsulated rejuvenators. Rejuvenators are encapsulated in the connected hollow pipe. And this method offers an advantage over capsule-based systems because the fibres increase the probability that rejuvenator will be released into cracks, which are more likely to pass through the fibre network [113]. Furthermore, the continuous pipe structure of the hollow system allows for the continuous supply of large volumes of rejuvenators, and the diameter of the fibres is usually 0.5–1.5 mm [114]. Zhang et al. prepared polyvinylidene fluoride resin (PVDF) hollow fibres by a one-step wet-spinning technology; the process is shown in Figure 9. They are distributed evenly in the bitumen base material and aim to reverse the ageing of bitumen and improve its crack repair capability [115]. The fibres were added to aged bitumen and still kept their integrality state by XCT test, proving they can resist the thermal actions of temperature changes in bitumen. Hollow fibres can survive in bitumen safely without debonding [114]. Guo et al. tested the self-healing ability and efficiency of bitumen ductility specimens with fibres and found that the self-healing rate reached 64% in the sample, which was larger than the pure bitumen sample. But an intersection angle between the tensile direction and the fibre hinders the flow of rejuvenator into the crack, which results in a poor self-healing effect [116].

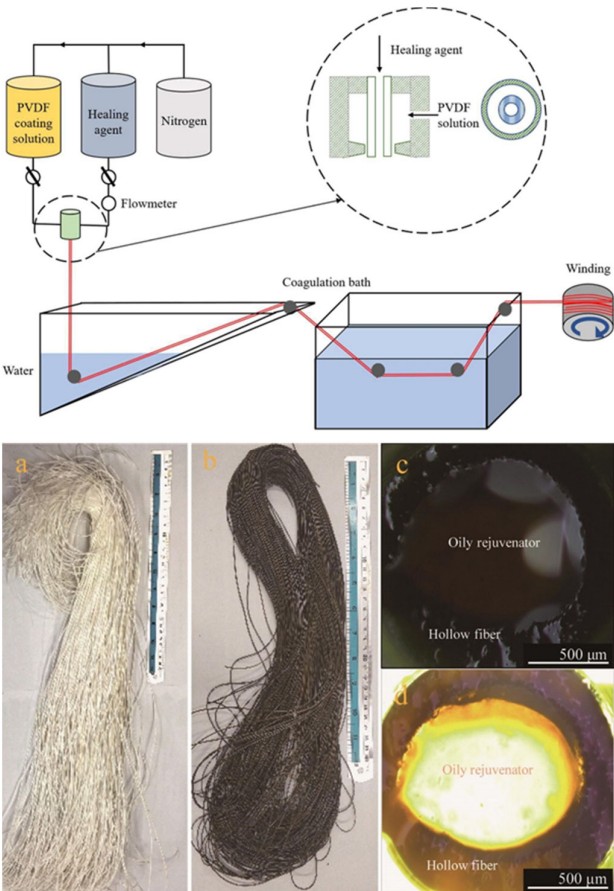

**Figure 9.** Schematic representation of the preparation of vascular hollow fibers and finished product: (**a**) fiber without rejuvenator, (**b**) fiber with rejuvenator, (**c**,**d**) cross-sectional of fiber with and without rejuvenator [114].

### 4.4.5. Compartment Fibers

Compartment fibres have their own advantages over hollow fibres. It does not allow the full release of the internal rejuvenator at one point of fracture, causing excessive local softening. Because the internal rejuvenator is separated into small droplets in the fibre. Its self-healing mechanism can be viewed as a merger of capsules and fibres. Tabakovi et al. used the physico-mechanical technique of wet-spinning to prepare the calcium alginate compartment fibres. The fibres have good thermal and mechanical properties because they still maintain their integrity after heating and mixing with bitumen [117]. These fibres present the rejuvenator distributed as individual droplets along their axis. The results demonstrated that the mechanical strength of mastic asphalt with fibres can increase by 36% compared to the reference, and the local micro-crack healing ability of samples with fibre also increases. After that, a microfluidic device was used to produce compartmented fibres with a commercial rejuvenator as a core and a sodium-alginate solution as a shell in Shu et al.'s study; the process is shown in Figure 10. The shell of fibres has excellent thermal stability and mechanical properties. The fibres were still intact after mixing and compacting with asphalt. The self-healing ratio of asphalt mix containing fibres increased by nearly 32% compared to asphalt mix without fibres. The system worked and enhanced the self-healing properties of the asphalt mixture [118,119]. In agent-encapsulated technologies, although the shell materials (epoxy resin, MF, MMF, Ca-alginate, and PVDF) used are all organic matter, they are basically non-toxic and will not flow into the environment and cause pollution when mixed into the asphalt. The core materials (commercial rejuvenator, sunflower oil, and waste oil) used are non-toxic and environment-friendly and

will be part of bitumen, which can be recycled. This further advances the sustainability of asphalt pavement.

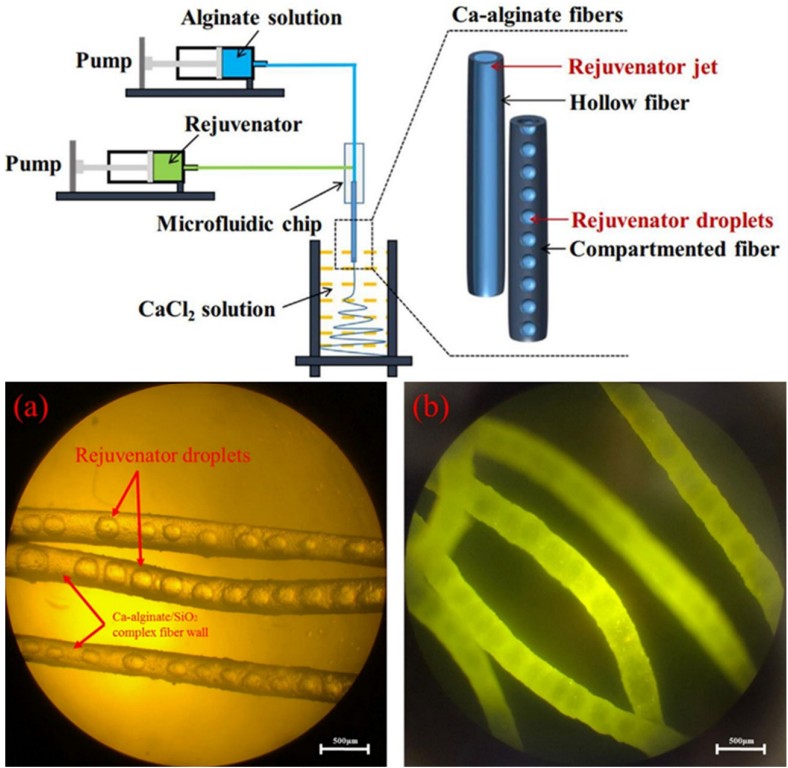

**Figure 10.** Microfluid synthetics compartment fibers and its morphology image, (**a**) optical microscope, (**b**) fluorescence microscope [118].

*4.5. Other Technologies*

There are also other technologies under study to help repair cracks in asphalt pavement. The comprehensive application of heating technology and rejuvenator supply technology is considered to overcome the shortcomings of their separate use. Xu et al. added both steel fibres and calcium alginate capsules to porous asphalt mixtures. The induction heating not only repairs cracks but also accelerates the outflow of the rejuvenators in the capsule and the diffusion in the bitumen [120]. Wan et al. design novel Ca-alginate capsules containing $Fe_3O_4$ powder, which can be induced to be heated to damage the shell of the capsules, leading to the rejuvenator's release by the low-frequency (2.45 GHz) microwave. So the capsule can achieve an artificially controlled release time [108,121,122]. Electrically conductive asphalt pavements are also designed to heal cracks in asphalt pavements by adding nanostructured conductive polymers, although asphalt is not electrically conductive [123]. At the crack, the resistance of the conductive asphalt will increase, and the temperature will also increase to heal cracks, according to Joule's law [124]. Three-dimensional printing technologies can also be applied for the maintenance of asphalt pavement cracks. Firstly, building 3D digital models of cracks by an advanced pavement distress detection system, then printing the crack directly in situ with prepared printing materials and equipment, and finally, checking printing quality by ultrasonic testing [125]. Jackson et al. realise bitumen can be directly used as a printing material to print into the cracks in the road surface to repair cracks at high temperatures to prolong the life of asphalt pavement [126]. Bitumen modified by polymer (SBS, Gilsonite, HDPE and crumb rubber in Lv et al. [127], SBS, HDPE and crumb rubber in Zhou et al. [128] crumb rubber, PPA, PE and gilsonite in Huang et al. [129]), nanomaterials (organoclay in Tabatabaee et al. [130], nano-silica in Ganjei et al. [131]), ionomers [132] and shape memory materials [133] to increasing the

self-healing properties of bitumen itself can also improve the crack healing ability of the asphalt pavement.

## 5. Conclusions

Crack-healing technology can effectively control pavement distress at an early stage, which can effectively extend the service life of asphalt pavement, reduce emissions, and save costs. This review introduces crack healing theories and evaluation methods and focuses on several crack healing techniques that are currently being applied and researched.The healing theories (the molecular diffusion healing model, phase field healing theory, surface energy healing theory, and capillary flow healing theory) cited by the researchers can explain the corresponding crack healing. However, in the actual healing process, crack healing is formed by the coupling of multiple healing models. Therefore, more attention should be paid to the advancement of the crack healing theory in the following research so that it can assist quantitative calculations and promote the application of crack healing technology.Hot pouring and fog sealing are relatively well studied and are the crack-healing techniques currently used in practical engineering. They mainly fill cracks from the outside, which can effectively prevent further damage to the asphalt pavement. However, when they are used, the cracks have generally developed into the middle and late stages, and their extension of the life of the asphalt pavement is relatively limited.Induction and microwave heating technologies have demonstrated significant efficacy in enhancing the crack healing capability of asphalt pavement, particularly in addressing microcracks. Extensive laboratory testing and some field test sections have been conducted, and they are now awaiting industry endorsement for promotion.The agents encapsulated in the technology not only heal cracks but also rejuvenate the aged asphalt pavement. Various encapsulation methods (saturated porous aggregates that encapsulate rejuvenators, core-shell polymeric microcapsules, ca-alginate capsules, hollow fibres, and compartment fibres) have been investigated in the laboratory. In order to promote the industrial application, more field test sections and large industrial mixing and compaction equipment applications need to be implemented. The comprehensive application of heating technology and rejuvenator supply technology, electrical conductivity asphalt pavements, 3D printing technologies, modified bitumen, and so on were also designed to repair cracks in asphalt pavement.

**Funding:** This research received no external funding.

**Conflicts of Interest:** The authors declare no conflict of interest.

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
