# Peer review of "A Methodological Review on Development of Crack Healing Technologies of Asphalt Pavement"

_sustainability, doi:10.3390/su15129659_

Round 1

Reviewer 1 Report

The submitted publication is a Review article. In my opinion, citations are missing in many places. There are many statements in the paper that are not supported by literature data, are these statements by the author?

In several places, according to the reviewer, the mechanism of "Crack healing" is not described at all, but rather the regular method of repairing the pavement.

Main comments

Points 4.1 and 4.2 the reviewer considers that these are not "Crack healing technologies" and are typical repair technologies. This is also indicated by the fact that it is not possible to apply the tests described in Table 1.

Section 4.3 no reference

The paper highlights fatigue cracking, what about low temperature cracking?

Minor

All equations, describe the components of the equation

Line 43-44 but what kind of cracks? fatigue? reflected? low temperature? need to be specified

Line 119 big space between by and AFM

Line 129, 140 should be below equations

Line 173 Reference in the tablce maybe with name of authors

Line 173 part of reference missing (number 31?)

Line 237 Big spaces between sentences?

Line 279 publication is from 2012, statement that further on the pavement is in good condition is a misuse

Line 282 space between 400 and m unit

Line 282 - 284 References needed

Line 365 Error

Line 458 (Figure 7) inverted figure

Line 478 (Figure 8) in which test fatigue was tested, divide figure into parts a) and b) change caption

Line 503 Missing in figure caption: a,b,c,d

Line 501 - 504 Consider splitting into two figures

Line 523 missing figure caption

Line 540 - 543 this section needs clarification, what do you mean by 3D printing, describe the technology in detail

Line 544 - 546 it seems that there should be more references to support the thesis

Author Response

the modified draft see the attachment.

The submitted publication is a Review article. In my opinion, citations are missing in many places. There are many statements in the paper that are not supported by literature data, are these statements by the author?

More references are added, and there are indeed some author's own opinions and comments in the paper, which will also be indicated in the paper.

In several places, according to the reviewer, the mechanism of "Crack healing" is not described at all, but rather the regular method of repairing the pavement.

The paper mainly focused on crack healing technologies, and the healing mechanisms they apply are familiar, and more mechanism explanations are added in each technical introduction.

 Main comments

Points 4.1 and 4.2 the reviewer considers that these are not "Crack healing technologies" and are typical repair technologies. This is also indicated by the fact that it is not possible to apply the tests described in Table 1.

It is also mentioned in the paper that these two technologies are mainly for repairing cracks and are the most widely used technologies today. It was therefore also added to the paper by the authors. The 3 points bending and SCB test in Table 1 sometimes are to evaluate the bonding performance of emulsified asphalt or other binders used in these two technologies, liking reference 31.

Section 4.3 no reference

Thanks, more references are added.

The paper highlights fatigue cracking, what about low temperature cracking?

Many researchers recommend the application of heating technology in spring, which is also considered to repair low-temperature cracks. At the same time, in the capsule healing technology, low-temperature cracks will also induce capsule rupture and low temperature crack healing. Related instructions will also be added to the paper.

 Minor

All equations, describe the components of the equation.

Thanks, has been added.

Line 43-44 but what kind of cracks? fatigue? reflected? low temperature? need to be specified

Here cracks contain all kinds of cracks.  Detailed explanation is added.

Line 119 big space between by and AFM

Has been deleted.

Line 129, 140 should be below equations

Has been modified.

Line 173 Reference in the table maybe with name of authors

Thanks, has been added.

Line 173 part of reference missing (number 31?)

Thanks, the format of the table is wrong, it has been added.

Line 237 Big spaces between sentences?

Has been deleted.

Line 279 publication is from 2012, statement that further on the pavement is in good condition is a misuse

Thanks, the reference just explains the paving of the test section. The pavement is good now is the latest information obtained by the authors, the citation position has been adjusted.

Line 282 space between 400 and m unit

Has been added.

Line 282 - 284 References needed

Has been added.

Line 365 Error

Has been modified.

Line 458 (Figure 7) inverted figure

Has been adjusted.

Line 478 (Figure 8) in which test fatigue was tested, divide figure into parts a) and b) change caption

has been added.

Line 503 Missing in figure caption: a,b,c,d

Has been added.

Line 501 - 504 Consider splitting into two figures

That is from preparation to the product, the picture has been merged.

Line 523 missing figure caption

Same to previous figure, the picture has been merged.

Line 540 - 543 this section needs clarification, what do you mean by 3D printing, describe the technology in detail

A detailed procedure for 3D printing to repair cracks has been added.

Line 544 - 546 it seems that there should be more references to support the thesis

Has been added.

Reviewer 2 Report

Review report A methodological review on the development of crack healing technologies of asphalt pavement

Since the publication is on Sustainability, I expect to read more discussions regarding resource usage and environmental damage related to road asphalting. I think it should be a separate paragraph (currently it is a part of the first paragraph). The abovementioned paragraph should be ended with a statement that cracks healing will mitigate most of these problems.

Since some reader is expected to be policy maker, not engineers, adding a paragraph that explain asphalt pavement for this audience is recommended. Including the aging of asphalt pavement and how water penetration deteriorates asphalt pavement is also necessary.

Line 130 -134, the notation should be with subscripts where necessary.

Table 1. Isn’t the “area between the modulus…” the energy difference? It is not the modulus but the strain-stress curve. The “dynamic modulus” needs to be defined for this proposal. Is it under quasi-static loading, or calculated from the speed of sound in the material? Because of the similarity, consider rearranging to have the fatigue-rest-fatigue test and DSR sweep test one after the other.

Line 192. It seems as if misuse of the word occurs.

Lines 195-196 maintain or extend?

Lines 214-219 it is not clear what the advantage of using sand in the mixture is.

Lines 221-226 need rephrasing

Subsection 4.3.1 Aren’t steel fiber susceptible to corrosion with time?

When reading about radiation heating I thought, how can it be that carbon black is not included? It is a cheap corrosion-resistant conductor. A fast search found that I’m not the first to consider it. See https://doi.org/10.1016/j.conbuildmat.2018.04.002 for example.

As the paper is submitted to sustainability, a short paragraph that compares the different inclusions to the asphalt from a sustainability perspective is needed.

Error! Reference source not found in line 365

4.4.1 What about the hazard of oil leaching to the environment? Is the oil has some toxicity?

Lines 399-402. Use subscript for the 3 in calcium carbonate. How long is long service time? Please supply a unit and a number.

Figure 7. The figure is mirrored

For each material used for healing, please indicate if it is toxic/nontoxic and if it is biodegradable. 

In the abstract, introduction, and subsection 4.3.1 the English is mostly understandable but very poor. My experience is that while AI cannot write a good scientific review, they are very good at rephrasing and improving the grammar if asked to do it a paragraph or two each time. Try it.

Author Response

Comments and Suggestions for Authors

Review report A methodological review on the development of crack healing technologies of asphalt pavement

Since the publication is on Sustainability, I expect to read more discussions regarding resource usage and environmental damage related to road asphalting. I think it should be a separate paragraph (currently it is a part of the first paragraph). The abovementioned paragraph should be ended with a statement that cracks healing will mitigate most of these problems.

Thanks, has been added.

Since some reader is expected to be policy maker, not engineers, adding a paragraph that explain asphalt pavement for this audience is recommended. Including the aging of asphalt pavement and how water penetration deteriorates asphalt pavement is also necessary.

Has been added.

Line 130 -134, the notation should be with subscripts where necessary.

Has been modified.

Table 1. Isn’t the “area between the modulus…” the energy difference? It is not the modulus but the strain-stress curve. The “dynamic modulus” needs to be defined for this proposal. Is it under quasi-static loading, or calculated from the speed of sound in the material? Because of the similarity, consider rearranging to have the fatigue-rest-fatigue test and DSR sweep test one after the other.

Thanks, it should be the area between the curves of the modulus versus the number of load cycles and the line of ½ modulus, it has been modified.

Line 192. It seems as if misuse of the word occurs.

Has been modified.

Lines 195-196 maintain or extend?

Has been modified.

Lines 214-219 it is not clear what the advantage of using sand in the mixture is.

It has been mentioned at previous sentence that the sand improves the skid resistance performance of road surface.

Lines 221-226 need rephrasing

It has been rephrased.

Subsection 4.3.1 Aren’t steel fiber susceptible to corrosion with time?

No, the steel fiber is completely wrapped by bitumen, it is difficult to contact with the air, so it will not corrode, has been added to the paper.

When reading about radiation heating I thought, how can it be that carbon black is not included? It is a cheap corrosion-resistant conductor. A fast search found that I’m not the first to consider it. See https://doi.org/10.1016/j.conbuildmat.2018.04.002 for example.

Thanks for your supplementary, carbon black is one of microwave absorbing materials, it has been added into section 4.3.2.

As the paper is submitted to sustainability, a short paragraph that compares the different inclusions to the asphalt from a sustainability perspective is needed.

Has been added.

Error! Reference source not found in line 365

Has been added.

4.4.1 What about the hazard of oil leaching to the environment? Is the oil has some toxicity?

The oil used in the asphalt rejuvenator is basically environmentally friendly and non-toxic, and has been added.

Lines 399-402. Use subscript for the 3 in calcium carbonate. How long is long service time? Please supply a unit and a number.

Has been modified.

Figure 7. The figure is mirrored

 Has been modified.

For each material used for healing, please indicate if it is toxic/nontoxic and if it is biodegradable.

has added.

Comments on the Quality of English Language

In the abstract, introduction, and subsection 4.3.1 the English is mostly understandable but very poor. My experience is that while AI cannot write a good scientific review, they are very good at rephrasing and improving the grammar if asked to do it a paragraph or two each time. Try it.

Thanks for your recommendation.

Round 2

Reviewer 1 Report

Thank you for making the corrections